# Non-Linear Gait Dynamics Are Affected by Commonly Occurring Outdoor Surfaces and Sex in Healthy Adults

**DOI:** 10.3390/s25134191

**Published:** 2025-07-05

**Authors:** Jill Emmerzaal, Patrick Ippersiel, Philippe C. Dixon

**Affiliations:** 1Department of Kinesiology and Physical Activity, McGill University, Montreal, QC H2W 1S4, Canada; jill.emmerzaal@mcgill.ca; 2School of Kinesiology and Physical Activity Sciences, University of Montreal, Montreal, QC H3T 1J4, Canada; patrick.ippersiel@umontreal.ca; 3Research Center of the CHU Sainte-Justine, Montreal, QC H3T 1C5, Canada

**Keywords:** biomechanics, gait dynamics, non-linear dynamics, surface effects, outdoor, sex differences, trunk accelerometry

## Abstract

(1) Background: Human walking involves adapting to diverse terrains, influencing gait biomechanics. This study examined how seven outdoor surfaces—flat–even, banked-right/-left, cobblestone, grass, sloped-down, and sloped-up—affect nonlinear gait dynamics in 30 healthy adults (14 females and 15 males). (2) Methods: Trunk and shank accelerations were analyzed for movement predictability (sample entropy, SE), smoothness (log dimensionless jerk, LDLJ), symmetry (step/stride regularity), and stability (short-/long-term Lyapunov exponents, LyE_*s*_, LyE_*l*_). (3) Results: Surface type significantly influenced all gait metrics, regardless of sex. Banked-right and sloped-down walking reduced SE, indicating less predictable movements. All surfaces except flat–even increased LDLJ, suggesting reduced smoothness. Cobblestone and sloped-down surfaces impaired step symmetry, while banked surfaces enhanced stride symmetry. LyE_*s*_ decreased on cobblestones (lower variability), while sloped-up increased it. LyE_*l*_ rose on all surfaces except cobblestones, indicating a more chaotic gait. No significant sex differences were found, though males showed a non-significant trend toward lower LyE_*s*_. Notably, sex–surface interactions emerged for SE and stride symmetry on banked-right surfaces, with females showing decreased SE and increased symmetry. (4) Conclusions: These findings underscore the importance of terrain and sex in gait dynamics research.

## 1. Introduction

Human bipedal walking requires navigating various terrains, impacting biomechanics and gait dynamics. The impact of surfaces on biomechanics is of interest to the clinical and research communities, in part due to potential implications for injury prevention and rehabilitation [1,2]. Past work reported gait adaptations (e.g., decreased dynamic stability, complexity, and smoothness) in response to sloped and irregular surfaces [2,3,4], suggesting that challenging surfaces may increase fall or injury risk in certain populations. Most gait research, however, is confined to gait laboratories with smooth floors and limited walking areas [1,3,4,5,6,7,8,9], limiting ecological validity and real-world applicability [10].

Wearable sensor technology (e.g., inertial measurement units (IMUs)) is bridging the gap between lab and real world research [11,12,13,14]. From trunk acceleration, (non-linear) gait dynamic metrics such as stability, symmetry, and smoothness have been quantified in walking, running, and posture biomechanics [1,4,5,13,15,16,17,18]. Unlike linear techniques (e.g., joint kinematics and kinetics), non-linear dynamics consider sequential observations within time-series and are suitable to reveal underlying patterns and relationships not immediately apparent with linear methods [19]. These metrics are sensitive to subtle changes in movement patterns and linked to injury, fall risk, and disease progression [1,20].

Moreover, there is a concerning sex data gap in research [21]. For example, in sports and exercise sciences, females only represent 34% of the studied population [22], highlighting the under-representation of females [21,22]. Therefore, a particular focus should lie in investigating female participants separate from their male counterparts. In biomechanical research, there have been some notable studies investigating the differences between females and males in gait [23,24,25,26] (add some reference). With regard to accelerometry-based gait research, significant between-sex differences in the spatio-temporal parameters of walking (e.g., walking speed, stride length, caudal-cranial activity) have been found [25], showing the need to explore potential sex differences in gait dynamics on outdoor surfaces. Moreover, previous work found that females were better able to attenuate accelerations from pelvis to shoulder and head levels in the medio-lateral direction, possibly indicating a better control strategy [26]. Whether we can see similar differences in non-linear dynamics is still underexplored.

Using IMUs gives us the opportunity to capture the demands of outdoor walking. Nonetheless, the impact of surfaces and sex on gait dynamics remains unclear. Furthermore, it is still unclear if a minimalistic sensor approach combined with non-linear dynamics can capture changes in walking dynamics due to surfaces. With this work, we hope to find possible metrics that could be of interest for future work in more clinical populations. Additionally, this provides a preliminary idea about the demands of surfaces. Therefore, this study aims to (1) investigate the effect of surface on gait dynamics captured from lower trunk acceleration; (2) investigate the difference in gait dynamics between females and males irrespective of surface; and (3) investigate the interaction effect of surface and sex on gait dynamics. We hypothesize that (1) surface influences non-linear dynamics irrespective of surface; (2) that there is a difference between females and males; and (3) that there are differential effects of surface between females and males.

## 2. Materials and Methods

This article is a revised and expanded version of a conference paper entitled ”Effects of common outdoor surfaces and sex on gait dynamics in healthy adults assessed with a single accelerometer”, which wil be presented at ISB 2025 Congress, Stockholm 27–31 July 2025.

### 2.1. Participants

We leveraged an existing dataset that included 30 healthy young adults walking on various outdoor surfaces [11]. All information regarding inclusion/exclusion criteria and surface details can be found in the dataset paper by Luo et al. [11]. One female participant was removed due to faulty lower back sensor data for the banked-left surface, resulting in a dataset of 14 females and 15 males. Descriptive statistics showed no age differences, but females were significantly shorter, lighter, and had a lower BMI.

### 2.2. Data Collection

Participants were equipped with 5 IMUs in the lower extremities and lower back (MTw Awinda, Xsens, Enschede, The Netherlands). Data from the lower back and both shanks were extracted for the following seven surfaces: flat–even, banked-right, banked-left, cobblestone, grass, sloped-down, and sloped-up. Participants performed 6 walking trials on each surface. These surfaces were all on the university campus that the participant navigated in their daily life; thereby, we do not expect there to be any sort of training effect or familiarization during data collection.

### 2.3. Data Processing and Analysis

#### 2.3.1. Preprocessing

Luo et al. [11] collected data using MTw Awinda software (Xsens, Enschede, The Netherlands) at 100 Hz, synchronized across sensors, and exported to .csv files (one file per trial, per surface). Python3 was used for all analysis herein.

Gait cycles were determined with the angular velocity signal about the medio-lateral axis of the shank sensor [27]. In short, gyroscope data were filtered using a 4th order, zero-lag, low-pass Butterworth filter with a 3 Hz cut-off frequency. Mid-swing was detected as the prominent positive peak, and the heel strike was determined as the first minimum after the mid-swing. We visually inspected the found peaks per participant, per trial, and per surface. In the event of a missed/or misslabeled step, we manually adjusted the peak finder thresholds.

The first and last steps were removed to ensure steady-state walking [28] and IMU signals from the individual trials were concatenated into a long time-series [18,29]. The Euclidean norm from 3D acceleration signals of the lower back was used for further analysis. As our metrics are sensitive to data length, we truncated each concatenated time-series to the least number of steps taken [30], resulting in a time-series of 80 steps per participant per surface. Gait dynamic measures were calculated from the concatenated acceleration signal without filtering.

#### 2.3.2. Non-Linear Gait Dynamics Metrics

Gait dynamics were evaluated in terms of (1) movement predictability, (2) movement smoothness, (3) movement symmetry, and (4) local dynamic stability and complexity.

Movement predictability was captured using sample entropy, with higher values indicating less periodicity and, thus, increased unpredictability/complexity. We used the algorithm by Richman and Moorman [31] with parameters from Yentes et al. [30] and Govindan et al. [32].

Movement smoothness was quantified with the log dimensionless jerk (LDLJ) using methods from Melendez-Calderon et al. [33]. Smoother movements have values closer to zero; they are often considered to be related to a higher skill level and are indicative of healthy neuro-muscular control [34].

Gait symmetry was determined using the dominant peaks in the unbiased autocorrelation, with the first and second dominant peak equivalent to the phase shift of one step and stride, respectively [17]. The height of the first two dominant peaks indicates the strength of the correlations between the steps and strides, respectively, with perfect symmetry equal to one.

Local dynamic stability and complexity were quantified using the maximum Lyapunov exponent (LyE) as the short-term (LyE_*s*_) and long-term (LyE_*l*_) exponents, respectively. State spaces were reconstructed using the false nearest neighbor algorithm and average mutual information theory [19,35]. From the state spaces, the short- and long-term LyE were estimated as the slope of the average logarithmic divergence of two initial nearest neighbors using Rosenstein et al. [36]’s method implemented in Python by Sarwar et al. [35] and slope boundaries determined by Bruijn et al. [37]. Higher values indicate lower dynamic stability and higher chaos [37,38].

### 2.4. Statistical Analysis

We used generalized estimating equations (GEE) [39] to analyze the effects of surface type, sex, and their interaction on our dependent variables: sample entropy, LDLJ, step symmetry, stride regularity, LyE_*s*_, and LyE_*l*_. GEE models were chosen because of the correlations between predictors, as measurements were taken from the same individual on the same day, and they are more robust with non-normally distributed data [39].

To test our hypotheses, we estimated the main effect of surface (aim 1) and sex (aim 2) using the following GEE formula: (1)$$\hat{y}={\mathrm{Surface}}_{\mathrm{ref}=\mathrm{flat}}+{\mathrm{Sex}}_{\mathrm{ref}=\mathrm{female}}+\mathrm{Height}+\mathrm{Weight},$$
where $\hat{y}$ is the dependent variable (e.g., sample entropy) and flat walking was set as the reference surface. This indicates that all other surfaces were compared to flat–even walking.

For the interaction effect of surface and sex (aim 3), we built a GEE model with an interaction term: (2)$$\hat{y}={\mathrm{Surface}}_{\mathrm{ref}=\mathrm{flat}}\times {\mathrm{Sex}}_{\mathrm{ref}=\mathrm{female}}+\mathrm{Height}+\mathrm{Weight}.$$
Here, $\hat{y}$ is the dependent variable, the flat surface is the reference surface, and female is set as the reference sex. We controlled for height and weight due to significant differences between our female and male groups.

Separate equations were used for each dependent variable, and the GEE models were fitted with an exchangeable correlation structure (assuming equal correlation within clusters) and Gamma family for non-normally distributed data, using Python’s statsmodels library [40]. To mitigate negative values, we added 1 to the LyE_*l*_ and used the inverse of the LDLJ. Residuals were checked and corrected with a log transform link function where needed.

Effect sizes were calculated using the following formula: (3)$$\mathrm{Effect}\;\mathrm{Size}=(e^{\beta }-1)\times 100,$$
where β is the coefficient, allowing interpretation in terms of percentage change.

## 3. Results

Model outcomes per dependent variable are reported in Table 1, Table 2 and Table 3 and summarized in Figure 1.

Significant surface effects for all gait dynamics variables (p≤0.05) were obtained on most surfaces, showing that walking surface impacts gait dynamics, independent of sex. Banked-right and sloped-down surfaces were associated with a significant decrease in sample entropy, reducing complexity. All surfaces significantly increased LDLJ compared to flat walking, indicating less smooth movements. The banked-right surface significantly increased step symmetry, while cobblestones and sloped-down surfaces decreased it. Banked-left and banked-right surfaces increased stride symmetry, while cobblestones, sloped-down, and sloped-up surfaces decreased stride symmetry. The cobblestone surface significantly decreased LyE_*s*_. Sloped-up walking was associated with a significant increase in LyE_*s*_, indicating that cobblestone surfaces decreased step-to-step variability whereas sloped-up walking increased step-to-step variability. All surfaces significantly increased LyE_*l*_, indicating increased complexity and chaos.

No significant main effects of sex on gait dynamics were found. Notably, a non-significant trend suggested a 13.7% decrease in step-to-step variability (LyE_*s*_) in males compared to females.

Significant surface–sex interactions were found for sample entropy and stride regularity on banked-right surfaces. Bank-right surface and being male significantly increased sample entropy. The female participants decreased their sample entropy by roughly 9.6% when walking on a banked-right surface, indicating a loss of complexity, whereas our male participants increased their sample entropy by approximately 0.5%.

For stride symmetry, the male group increased their stride symmetry by 0.88% compared to an increase in stride symmetry for our female participants of 3.77%. No other significant interaction effects were found. Detailed percentage changes are shown in Table 2 and Table 3.

## 4. Discussion

### 4.1. Surface Effects

Our first hypothesis was confirmed, showing that surface affects gait dynamics. Significant decreases in sample entropy during banked-right and sloped-down walking were observed, indicating a loss of complexity. For sloped-down walking, this contrasts with [3], who found increased complexity. These discrepancies are likely due to methodological differences, which precludes direct comparison. Vieira et al., [3] used marker-based data to capture the Lyapunov exponent and sample entropy from sternum velocity. Measures of sample entropy on continuous time-series have been found to be sensitive to input parameters [41]; therefore, changing the input signal might change the direction of the effect. Nevertheless, both studies conclude that sloped-down walking has a significant effect on sample entropy during gait.

Movement smoothness, quantified by LDLJ, was highly sensitive to surface, with all surfaces prompting a decrease in smoothness compared to flat walking. This corroborates previous studies investigating smoothness on an uneven brick surface [4]. Surprisingly, only cobblestones and sloped-up surfaces impacted dynamic stability, as calculated with the short-term Lyapunov exponent, partially corroborating [3] during incline treadmill walking. We found more differences in the long-term Lyapunov exponent compared to the short-term exponent, with all surfaces showing increased complexity and chaos. While absolute thresholds for clinical significance in non-linear dynamics are still under development, several studies provide compelling evidence that even modest changes in sample entropy, smoothness, and the Lyapunov exponent can reflect a meaningful alteration in gait dynamics, reflecting injury risk and fall risk [6,7,42,43,44,45]. Therefore, further work is needed to fully understand the dynamics of outdoor walking, with a focus on movement smoothness (LDLJ), stride-to-stride fluctuations and complexity (LyE_*l*_), and symmetry to assess the effects of walking surfaces on injury and fall risk in healthy and clinical populations.

### 4.2. Sex Effects

Contrary to our second hypothesis, no statistically significant between-sex differences were found, independent of walking surface. We observed a non-significant reduction of 13% in LyE_*s*_ for males, independent of walking surface. Previous work has linked the LyE_*s*_ to fall risk and knee osteoarthritis disease state [18,37]. Bruijn et al. [37] showed that people with a higher fall risk show higher values of the LyE_*s*_, whereas Emmerzaal et al. [18] found that people with knee osteoarthritis had lower values compared to healthy controls. These seemingly opposite results might indicate that different diseases or clinical groups might have an unique movement profile. As the male and female group in this study are healthy, young individuals, there is no reason the believe that these differences in LyE_*s*_, are the results of an underlying disease state. Therefore, we cannot give any clinical meaning to these results, besides that females and males might have a different dynamic stability strategies in terms of the LyE_*s*_.

### 4.3. Surface–Sex Effects

Our third hypothesis was confirmed, with interactions observed for some metrics. The significant interaction between the banked-right surface and sex for sample entropy (decreased for females and increased for males) suggests that females (vs males) respond to this change with adaptations linked to a less (vs more) adaptable gait pattern with respect to performing step-to-step adjustments to regulate balance control [44,45]. Further indication that females adopt a more predictable and regular movement pattern is in the interaction between banked-right surface and sex for stride symmetry. Females increased and males decreased their stride symmetry, suggesting that females (vs males) adopt a more (vs less) regular movement pattern. Moreover, the interaction between sloped-down walking and sex, although not statistically significant, suggests a practical difference in how sloped-down walking affects LyE_*s*_ between males and females. The narrow scope (i.e., only on banked-right surfaces) of significant interactions found may reflect insufficient power instead of a genuine absence of interactions. Therefore, future studies with larger sample sizes are warranted to explore potential sex-specific gait adaptations.

### 4.4. Limitations

This study has limitations that warrant discussion. First, walking speed was not controlled, which may influence gait dynamics; however, we mitigated this by using a consistent step count per participant across surfaces. Second, as our metrics are calculated on unfiltered data, there might be an influence of signal noise, sensor drift, or environmental interference. To help mitigate this, data collection procedures were standardized (e.g., sensor placement, sensor calibration) and short-duration trials were collected in a controlled outdoor environment. Third, while we observed sex interaction effects, identifying their origin was not a primary aim of the dataset, making post hoc in-depth analysis difficult. Future research should explore whether these differences are inherent to sex or influenced by other factors. For example, breast motion is known to alter pelvis, trunk, and lower limb kinematics during exercise [46,47,48], thus potentially driving non-linear dynamics that we observed herein measured at the lower trunk level. Fourth, trials were concatenated into a single time-series, which may introduce variance; however, since this method was applied uniformly, with visually inspected step detection and concatenated results, we believe its impact on conclusions may be minimal. Fourth, we used a step detection algorithm based on the angular velocity of the shank sensor. We visually inspected the step detection outcomes and adjusted thresholds where needed, and future work should formally investigate the robustness of step detection algorithms in daily life. Fifth, the sample was small and consisted of university-aged young adults, limiting generalizability to other populations and limiting the power of this study. Furthermore, there could be slight bias caused by ordering effects, gait adaptation to surface, or fatigue; however, we sampled from a young, healthy population, and walking was performed on terrain that they were meant to encounter in their everyday life. Therefore, we believe that biases introduced by ordering effects, gait adaptation, or fatigue are minimal in this study. Finally, some participants (primarily male) could be classified as overweight via the BMI metric. We controlled for height and weight to help minimize bias; however, future work should investigate the impact of body composition and fitness level on outdoor walking.

## 5. Conclusions

This study investigated the impact of various outdoor surfaces on gait dynamics and explored potential sex differences in these dynamics. Even though all participants were able to navigate these surfaces with success (e.g., no falls, or disruption of gait), all conditions, especially banked-right, cobblestones, and sloped-down, induced changes in non-linear walking dynamics, as derived from lower trunk accelerations, compared to flat walking. These surfaces prompted a deterioration in movement smoothness, altered symmetry, increased chaos, and more stride-to-stride fluctuation, suggesting the need for a higher level of motor skill [15,34,38].

Our results highlight the need for more data collection in outdoor setting to fully capture the demands of various surfaces on the gait dynamics of females and males. Greater consideration of sex differences in gait studies is necessary to enhance understanding of movement adaptation strategies. Future research should explore the physiological implications of these findings and extend analyses to older/clinical populations and use longer trials so concatenation of signals will not be necessary.

## Figures and Tables

**Figure 1 sensors-25-04191-f001:**
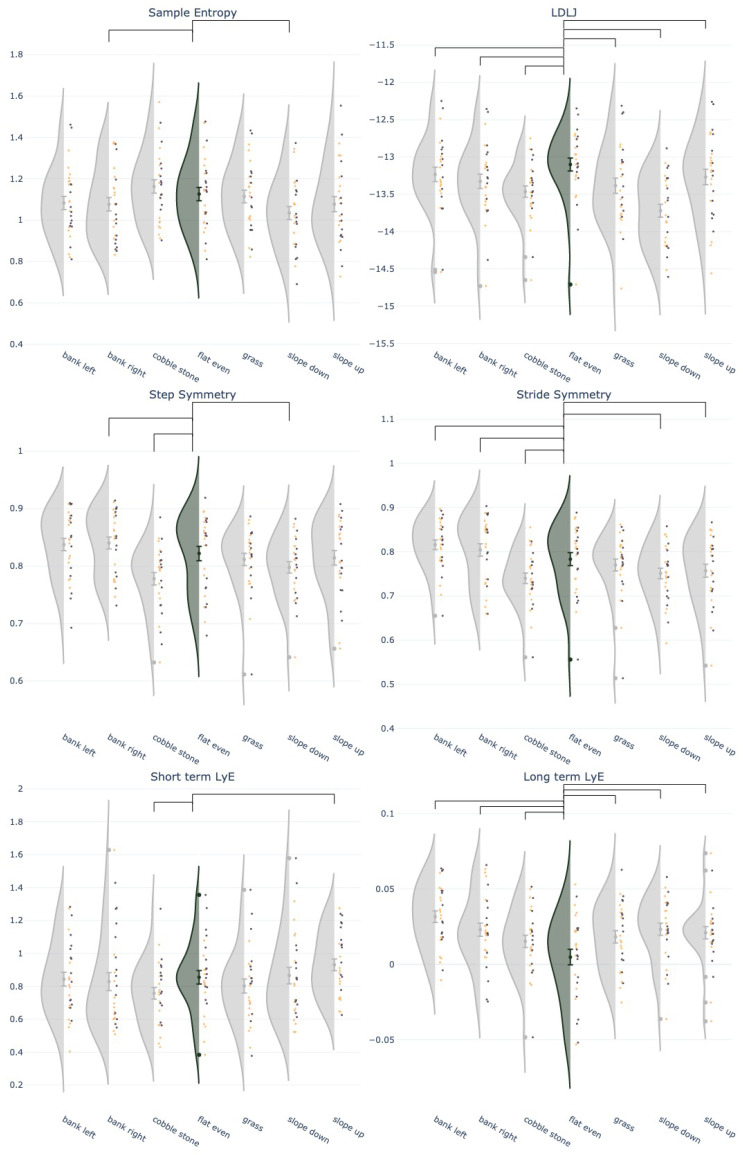
Raincloud plots of the gait dynamic variables (dimensionless quantities) on the different surfaces compared to the reference surface, flat–even in forest green/darker. Each dot represents an individual (purple/dark = female and yellow/light = male) and the half-violin plots show the projected distribution. Mean ± 95% confidence intervals are displayed. All surfaces are compared to the flat walking surface. Significant differences (p≤0.05) with flat–even walking are indicated by the black bars above the graph.

**Table 1 sensors-25-04191-t001:** Model outcomes for the main effects of surface regardless of sex, and sex regardless of surface. β-Coefficients, *p*-values, and percentage difference (%) from GEE regression analysis for all dependent variables: sample entropy, log dimensioness jerk (LDLJ), step symmetry, stride symmetry, short-term Lyapunov exponent (LyE), and long-term LyE. Significant *p*-values are in bold.

	Sample Entropy			LDLJ			Step Symmetry			Stride Symmetry			Short-Term LyE			Long-Term LyE		
	β	p	%	β	p	%	β	p	%	β	p	%	β	p	%	β	p	%
Main surface effect																		
FL-BL	−0.04	0.14	−3.87	0.01	**0.01**	1.01	0.02	0.07	1.92	0.04	**0.00**	4.16	−0.01	0.84	−1.13	0.03	**0.00**	2.67
FL-BR	−0.05	**0.04**	−4.40	0.02	**0.00**	1.71	0.02	**0.01**	2.27	0.03	**0.04**	2.63	−0.03	0.57	−3.42	0.02	**0.01**	1.81
FL-CS	0.03	0.10	3.28	0.03	**0.00**	2.78	−0.05	**0.00**	−5.30	−0.06	**0.00**	−5.52	−0.11	**0.01**	−10.60	0.01	0.06	1.04
FL-Gr	−0.01	0.65	−1.04	0.02	**0.00**	2.16	−0.01	0.17	−1.21	−0.02	0.09	−1.66	−0.06	0.30	−5.99	0.01	**0.05**	1.34
FL-SD	−0.08	**0.00**	−8.12	0.05	**0.00**	4.73	−0.03	**0.00**	−2.85	−0.04	**0.00**	−4.08	0.01	0.81	1.21	0.02	**0.00**	1.84
FL-SU	−0.04	0.12	−4.31	0.01	**0.02**	1.28	−0.01	0.49	−0.84	−0.03	**0.03**	−3.27	0.09	**0.03**	9.77	0.02	**0.00**	1.61
Main sex effect																		
Sex	−0.03	0.72	−3.11	−0.01	0.55	−1.22	0.02	0.66	1.69	0.03	0.53	2.80	−0.15	0.18	−13.71	0.01	0.21	0.77

Abbreviations: FL = flat–even (serves as the reference surface). BL = banked-left, BR = banked-right, CS = cobblestone, Gr = grass, SD = sloped-down, and SU = sloped-up. β = coefficient, *p* = significance value, and % = percent difference.

**Table 2 sensors-25-04191-t002:** Percentage difference for the female and male group outcomes for the interaction effect of surface and sex from GEE regression analysis for our various dependent variables: sample entropy, log dimensionless jerk (LDLJ), step symmetry, stride symmetry, short-term Lyapunov exponent (LyE), and long-term LyE. Significant *p*-values are in bold.

		Sample Entropy		LDLJ		Step Symmetry		Stride Symmetry		Short-Term LyE		Long-Term LyE	
		%	p	%	p	%	p	%	p	%	p	%	p
FL-BL	Female	−4.96	0.68	1.12	0.77	2.44	0.65	5.59	0.32	−5.04	0.48	3.80	0.10
	Male	−2.81		0.90		1.45		2.85		2.73		1.63	
FL-BR	Female	−9.58	**0.01**	1.53	0.62	3.77	0.10	6.02	**0.01**	−4.29	0.88	2.10	0.67
	Male	0.59		1.88		0.88		−0.45		−2.56		1.55	
FL-CS	Female	3.90	0.76	2.86	0.81	−4.88	0.60	−5.26	0.81	−13.58	0.41	1.67	0.27
	Male	2.69		2.70		−5.69		−5.75		−7.67		0.47	
FL-Gr	Female	−1.61	0.80	1.28	0.10	−1.00	0.82	−1.06	0.56	−8.13	0.71	2.60	0.06
	Male	−0.49		2.98		−1.40		−2.21		−3.89		0.18	
FL-SD	Female	−11.42	0.17	5.02	0.62	−1.76	0.24	−2.66	0.21	−4.97	0.21	2.65	0.17
	Male	−4.94		4.47		−3.84		−5.36		7.29		1.10	
FL-SU	Female	−5.38	0.70	1.32	0.95	−0.05	0.53	−2.11	0.47	7.40	0.62	2.29	0.19
	Male	−3.27		1.25		−1.56		−4.32		12.10		0.98	

Abbreviations: FL = flat–even (serves as the reference surface). BL = banked-left, BR = banked-right, CS = cobblestone, Gr = grass, SD = sloped-down, and SU = slope up. *p* = significance value and % = percent difference.

**Table 3 sensors-25-04191-t003:** β-coefficients, *p*-values, and percentage difference (%) from the interaction effect GEE regression analysis for our various dependent variables: sample entropy, log dimensionless jerk (LDLJ), step symmetry, stride symmetry, short-term Lyapunov exponent (short-term LyE), and long-term Lyapunov exponent (long-term LyE). The main surface effects are for the reference group (female) only. The interaction effects are the % differences compared to the reference group.

	Sample Entropy			LDLJ			Step Symmetry			Stride Symmetry			Short-Term LyE			Long-Term LyE		
	β	p	%	β	p	%	β	p	%	β	p	%	β	p	%	β	p	%
Surface effect																		
FL-BL	−0.05	0.15	−4.96	0.01	0.08	1.12	0.02	0.14	2.44	0.05	0.02	5.59	−0.05	0.32	−5.04	0.04	0.00	3.80
FL-BR	−0.10	0.00	−9.58	0.02	0.00	1.53	0.04	0.01	3.77	0.06	0.01	6.02	−0.04	0.55	−4.29	0.02	0.03	2.10
FL-CS	0.04	0.02	3.90	0.03	0.00	2.86	−0.05	0.00	−4.88	−0.05	0.00	−5.26	−0.15	0.00	−13.58	0.02	0.01	1.67
FL-Gr	−0.02	0.47	−1.61	0.01	0.13	1.28	−0.01	0.50	−1.00	−0.01	0.54	−1.06	−0.08	0.27	−8.13	0.03	0.00	2.60
FL-SD	−0.12	0.00	−11.42	0.05	0.00	5.02	−0.02	0.25	−1.76	−0.03	0.15	−2.66	−0.05	0.44	−4.97	0.03	0.00	2.65
FL-SU	−0.06	0.19	−5.38	0.01	0.08	1.32	0.00	0.98	−0.05	−0.02	0.40	−2.11	0.07	0.26	7.40	0.02	0.00	2.29
Sex effect (Male)																		
Sex	−0.06	0.53	−6.109	−0.01	0.49	−1.36	0.03	0.49	2.96	0.05	0.32	5.14	−0.20	0.16	−18.15	0.02	0.06	2.08
Surface:Sex(M) interaction																		
FL-BL:Sex	0.02	0.68	2.26	0.00	0.77	−0.22	−0.01	0.65	−0.97	−0.03	0.32	−2.59	0.08	0.48	8.17	−0.02	0.10	−2.10
FL-BR:Sex	0.11	0.01	11.25	0.00	0.62	0.35	−0.03	0.10	−2.78	−0.06	0.01	−6.10	0.02	0.88	1.81	−0.01	0.68	−0.54
FL-CS:Sex	−0.01	0.76	−1.17	0.00	0.81	−0.16	−0.01	0.60	−0.85	−0.01	0.81	−0.51	0.07	0.41	6.84	−0.01	0.27	−1.18
FL-GR:Sex	0.01	0.80	1.14	0.02	0.10	1.68	0.00	0.82	−0.40	−0.01	0.56	−1.16	0.05	0.71	4.61	−0.02	0.06	−2.35
FL-SD:Sex	0.07	0.17	7.31	−0.01	0.62	−0.52	−0.02	0.24	−2.12	−0.03	0.21	−2.78	0.12	0.21	12.90	−0.02	0.17	−1.52
FL-SU:Sex	0.02	0.70	2.23	0.00	0.95	−0.07	−0.02	0.53	−1.51	−0.02	0.47	−2.26	0.04	0.62	4.38	−0.01	0.19	−1.28

Abbreviations: FL = flat–even (serves as the reference surface). BL = banked-left, BR = banked-right, CS = cobblestone, Gr = grass, SD = sloped-down, and SU = sloped-up. *β* = Coefficient, *p* = significance value and % = percent difference.

## Data Availability

We used a publicly available data set from Luo et al. [11] and can be found at: https://doi.org/10.6084/m9.figshare.c.4892463.v1, URL (accessed on 15 January 2024). The code associated with the results of this study can be found at: https://github.com/mcgillmotionlab/non_linear_dynamics_outdoor_surfaces.git.

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
