# Peer review of "Non-Linear Gait Dynamics Are Affected by Commonly Occurring Outdoor Surfaces and Sex in Healthy Adults"

_sensors, 2025, doi:10.3390/s25134191_

Round 1

Reviewer 1 Report

Comments and Suggestions for Authors

This study demonstrates that terrain type has a significant impact on nonlinear gait dynamics in healthy adults, influencing movement predictability, smoothness, symmetry, and stability. Notably, uneven or inclined surfaces—such as cobblestones and slopes—reduce gait smoothness and predictability, whereas certain terrains, like banked paths, can enhance stride symmetry. Although no significant overall sex differences were found, some interaction effects between surface type and sex suggest nuanced biomechanical responses. These findings underscore the crucial role of environmental factors—particularly surface variation—in shaping human locomotion patterns and highlight the importance of incorporating terrain and sex considerations in future gait analysis and mobility research. The following recommendations aim to enhance this research study.

1. In the introduction, the claim that IMUs bridge the lab-to-field gap is asserted without acknowledging their limitations (e.g., sensitivity to sensor placement, signal noise, or drift). This unbalanced presentation overlooks potential technological drawbacks that could affect data reliability or interpretability. Please clarify this. 
2. Although the authors argue that participants were familiar with the surfaces, the design does not control for ordering effects, adaptation, or fatigue across the six trials per surface. This may introduce systematic bias.
3. Although surface × sex interactions were significant for sample entropy and stride regularity only on banked right surfaces, no other surface × sex interactions were found significant. This narrow scope may reflect insufficient statistical power rather than a genuine absence of interaction effects, especially since previous literature suggests sex-specific adaptations on uneven terrain.
4. The statistical results are not contextualized in terms of real-world implications. For example, does a 4.4% decrease in sample entropy or a 10.6% decrease in LyEₛ represent a clinically meaningful change in gait predictability or stability? This connection is essential for applied research in rehabilitation or fall prevention.
5. The contrast between this study and prior findings (e.g., reference [3] on sloped walking) is attributed to “methodological differences.” Still, no specific methodological variables are identified (e.g., slope gradient, sensor type, participant condition). This weak justification fails to engage meaningfully with conflicting literature, thereby weakening scholarly rigor.
6. Given the reliance on IMUs and unfiltered data for entropy and Lyapunov measures, signal noise, sensor drift, and environmental interference could substantially affect the integrity of nonlinear analysis. This is not discussed in the limitations despite being highly relevant to signal-derived metrics.
7. There are typographical and language errors. Errors such as “comunuties” instead of “communities,” “seperate” instead of “separate,” and inconsistent punctuation detract from the professionalism and readability of the section.

Reviewer 2 Report

Comments and Suggestions for Authors

This paper investigates at how walking on different outdoor surfaces, like flat ground, sloped paths, grass, cobblestones, and banked turns, affects gait patterns in healthy adults, especially when comparing men and women. The authors collected trunk and shank acceleration data from 30 participants (14 women, 15 men) and analyzed it using measures like sample entropy, dimensionless jerk, symmetry in step and stride, and Lyapunov exponents. While the findings are interesting, there are still some issues that need to be addressed:

  1. The acceleration data was processed using the Euclidean norm of the 3D signal, but was it filtered at all to remove noise, or was the raw signal used straight from the sensors?
  2. It doesn’t seem like walking speed was controlled. That could affect the results, though the authors tried to account for this by standardizing the number of steps.
  3. All trials were combined into a single time series for each person. That might introduce variability? Was this checked or considered?
  4. Heel strikes were identified using angular velocity from the shank. How well does that method hold up across different surfaces? Any chance steps were misidentified?
  5. For the stats, GEE was used to deal with repeated measures, but why choose an exchangeable correlation structure? Something like autoregressive might better reflect the structure of gait data.
  6. They used the Gamma family for non-normal data, which makes sense. But were residuals checked, or did they try any transformations to see if that improved model fit?
  7. Some percentage changes were reported (9.6% drop in sample entropy), but it’s not clear if those are clinically meaningful or just statistically significant.
  8. For the Lyapunov exponents, males showed a 13.7% lower value. How does that compare to what's been seen in fall-risk or clinical gait studies?
  9. There’s a brief mention that breast movement might be a confounder in women. Was anything done to account for that?
  10. Lastly, the paper makes a good case for studying gait in real world settings. Would future studies benefit from pairing IMUs with pressure insoles or video to back up the findings?

Round 2

Reviewer 1 Report

Comments and Suggestions for Authors

The authors have thoroughly responded to all previously provided feedback, and the revised manuscript meets the necessary standards for publication.